# High-throughput sequencing reveals endophytic bacterial differentiation of common truffles (*Tuber* spp.) in China: diversity, biogeographical patterns, and fungal health implications

Man Guo,[1,2,3] Zhilan Xia,[1] Xunyang He,[3,4] Shanping Wan,[5] Yanliang Wang,[2] Shaolin Fan,[5] Jesús Pérez-Moreno,[6] Zhenyan Yang,[2] Chengmo Yang,[2] Dong Liu,[2] Fuqiang Yu[2]

**ABSTRACT** As valuable hypogeous fungi, truffles depend on fruiting-body-associated microorganisms for lifecycle functions like growth and nutrient cycling. This study sampled fruiting bodies of 10 *Tuber* species from 16 sites across six major truffle-producing provinces in China, characterizing endophytic bacterial communities via high-throughput sequencing and multivariate analysis. Proteobacteria dominated the endophytic bacteria, with *Bradyrhizobium* as the prevalent genus. Significant genus-level compositional differences occurred across provenances and species: *Bradyrhizobium* reached 99.80% relative abundance in *Tuber sinense* from Mengzi, Yunnan, versus 7.90% in *Tuber shii* from Dali (12.6-fold difference). Shannon diversity indices ($n = 48$) revealed striking species- and altitude-related variations ($P < 0.001$): *Tuber lijiangense* (5.111) and *T. shii* (5.091) had the highest diversity, while *T. sinense* (1.336) had the lowest (3.8-fold gap). Subtropical Dali samples exhibited a sevenfold higher diversity compared to those from the Mengzi region, which is geographically closer to the tropics. Non-metric scaling and principal coordinates analysis identified environmental factors (soil, climate) and host species as primary drivers, with species effects potentially overriding environment. Five core taxa (all *Rhizobiales*) suggested nitrogen-fixing roles, while *Variovorax* (via linear discriminant analysis effect size) emerged as an external-disturbance-sensitive opportunist. This study clarifies endophytic bacterial variation patterns and drivers, identifies key taxa, and informs truffle ecological interactions, providing a scientific basis for sustainable resource management and conservation.

**IMPORTANCE** This study underscores the critical importance of truffle endophytic bacteria in mediating fungal health and ecological resilience, addressing a major knowledge gap in hypogeous fungal microbiome research. By integrating high-throughput sequencing across 10 *Tuber* species in China, we reveal how bacterial communities (dominated by *Bradyrhizobium*) shape biogeographical patterns and functional roles like nitrogen fixation. These findings advance understanding of microbe-fungal symbioses, with direct applications for sustainable truffle cultivation (e.g., microbial inoculants) and climate-resilient management—aligning perfectly with AEM's focus on applied microbial ecology and biotechnological relevance.

**KEYWORDS** truffles, endophytic bacteria, microbiome, diversity, *Bradyrhizobium*

The genus *Tuber* belongs to the family Tuberaceae within the phylum Ascomycota. Truffles typically produce edible hypogeous fruiting bodies (commonly referred to as "true truffles") with a protracted maturation period requiring up to 6 months

**Peer Reviewer** Feng Huang, Guangdong Academy of Agricultural Sciences, Guangzhou, Guangdong, China

Address correspondence to Dong Liu, liudongc@mail.kib.ac.cn, Xunyang He, hbhpjhn@isa.ac.cn, or Fuqiang Yu, fqyu@mail.kib.ac.cn.

The authors declare no conflict of interest.

for complete development (1). Certain *Tuber* species, renowned for their distinctive aromatic profiles, hold exceptional economic value globally—for instance, the Italian white truffle (*T. magnatum*) and Périgord black truffle (*T. melanosporum*) range from €600 to €6,000 per kilogram, while the summer truffle (*T. aestivum*) and Chinese black truffle (*T. indicum* complex) are also highly prized in culinary markets. China harbors significant truffle biodiversity (over 60 documented species), among which *T. indicum* exhibits wide geographical distribution (predominantly in southwestern and northeastern regions), high productivity, and morphological-genetic similarities to *T. melanosporum*, making it a promising candidate for commercial development (2, 3). Notably, truffle cultivation intersects environmental, economic, and social dimensions: ecologically, *Tuber* species establish essential ectomycorrhizal (ECM) symbioses with diverse plant families (Pinaceae, Fagaceae, Myrtaceae, Salicaceae), enhancing host nutrient acquisition, stress resilience, and root development to support forest community stability and biodiversity (4). Economically, truffle fruiting bodies serve as luxury ingredients and potential medicinal resources, while cultivation practices—originally developed in Spain, France, and Italy—have expanded to non-native regions like Australia and New Zealand (5). European countries have further adopted truffle cultivation as a strategy for land stabilization, reforestation, and rural economic development, solidifying its role as a long-term cash crop (6).

While significant progress has been made in understanding truffle evolution, genetic diversity, chemical composition, and cultivation, the biological intricacies of truffle fruiting bodies remain enigmatic. The complex life cycle of truffles initiates with haploid spore germination, progressing to free-living haploid mycelia in soil. Subsequent colonization of host roots establishes mutualistic symbiosis, culminating in fruiting body formation after 5–7 years (7). Throughout development, truffles engage in intricate microbial interactions within their soil environment. Fluorescence *in situ* hybridization studies have identified dominant bacterial populations within mature *T. melanosporum* ascomata (8, 9), while research by Le Roux (10) revealed an obligate symbiotic relationship between *Bradyrhizobiaceae* bacteria and *T. melanosporum* mycelia. These findings elucidate the complexity of truffle–microbe interactions, revealing relationships that extend beyond mere coexistence to encompass specialized functional partnerships critical for truffle ecology and development.

Fungi act as symbiotic hosts, frequently harboring microbial endosymbionts (11) and establishing cooperative relationships within their mycelial networks (12). They create unique ecological niches that selectively recruit specific bacterial taxa (13, 14). Diverse microbial communities inhabit macrofungi (15, 16), with certain members enhancing spore germination, mycelial health, and growth (16). Truffles represent complex microhabitats hosting diverse microbial consortia (17). Over recent decades, culture-dependent and 16S rRNA sequencing approaches have characterized microbial communities in commercial truffle species, including *T. borchii* (18), *T. aestivum* (19, 20), *T. magnatum* (21–23), and *T. melanosporum* (24, 25). These studies demonstrate that truffle ascomata harbor rich microbial diversity. Bacterial community analyses reveal *Pseudomonas*, *Bacteroides*, and Gram-positive bacteria as dominant cultivable taxa (18, 26), while Proteobacteria, Bacteroidetes, Firmicutes, and Actinobacteria constitute core phyla across all studied species through both culture-dependent and culture-independent methods (27). Emerging evidence confirms microbial involvement in ascomata formation (27, 28), with *Bradyrhizobium* species being particularly abundant in truffle ascomata (8, 29, 30). Filamentous fungi have also been observed within truffle ascocarps (28). As microbiome research advances, the critical roles of truffle-associated microbiota in growth regulation, aroma biosynthesis, nutrient acquisition, and defense mechanisms are increasingly recognized. Given that China harbors a rich diversity of *Tuber* species (more than 60 recorded to date) and that dominant taxa such as *T. indicum* possess substantial commercial potential, clarifying the diversity, biogeographic patterns, and driving factors of *Tuber*-associated endobacteria will provide a scientific basis for the

sustainable exploitation of these resources and for formulating microbial management strategies during artificial cultivation.

Current research predominantly prioritizes commercially valuable truffle species, leaving China's rich truffle diversity insufficiently explored. Furthermore, methodological limitations inherent in conventional approaches constrain comprehensive microbial community profiling. This study aims to address these gaps by conducting a systematic analysis of microbiome dynamics across multiple *Tuber* species and diverse geographical regions in China. We hypothesize that the endobacterial community divergence in *Tuber* is governed by (i) host‑species‑specific traits and (ii) geographic environmental gradients, with their interaction exerting a species‑dependent effect that ultimately assembles a functionally coherent core microbiome. To test this hypothesis, we employ high-throughput sequencing technology to conduct exhaustive microbial community analyses on truffle ascomata from various regions. This investigation seeks to provide novel ecological insights into truffle endophytic microbiology, aiming to delineate robust scientific foundations useful for conservation, sustainable utilization, and responsible promotion development of China's truffle industry.

## RESULTS

### ITS-based phylogeny and species delimitation of truffle samples

Phylogenetic analysis of the internal transcribed spacer (ITS) region from 16 truffles resolved 10 species: *Tuber sinense*, *T. indicum*, *T. variabilisporum*, *T. pseudohimalayense*, *T. shii*, *T. lijiangense*, *T. sinoaestivum*, *T. huidongense*, *T. cf. sinoalbidum*, and *T. formosanu*.

Specimen assignments were as follows: YmTe and YksTe to *T. sinense*; YjTi and GbTi to *T. indicum*; YslTv to *T. variabilisporum*; YdTp and YssTp to *T. pseudohimalayense*; YdTs to *T. shii*; YdTl to *T. lijiangense*; ShTa to *T. sinoaestivum*; YqTh to *T. huidongense*; YzTb to *T. cf. sinoalbidum*; and SbTf, LdTf, LaTf, and NhTf to *T. formosanum*.

All sequences exhibited bootstrap support ≥90% (1,000 replicates) relative to GenBank references (e.g., *T. indicum* AM932205 and *T. sinense* MF627968), and out-group accessions (*Choiromyces sichuanensis* MW380902 and OK585070) formed a well-supported clade, confirming the robustness of species identifications.

Phylogenetic topology revealed three notable patterns: (i) conspecific accessions (e.g., the four *T. formosanum* samples) formed monophyletic clades that were clearly delimited from sister taxa such as *T. indicum*; (ii) geographically proximate collections from Dali, Yunnan (YdTp, YdTs, YdTl), although assigned to different species, occupied adjacent branches, implying that geographic distance may influence lineage divergence; and (iii) *T. indicum* sequences were phylogenetically remote from the European sister species *T. melanosporum* (AF106875), consistent with previously reported morphological and genetic distinctions (Fig. 1).

### Endophytic bacterial community composition

This study identified a rich bacterial assemblage in truffle ascomata, encompassing 15 phyla, 30 classes, 82 orders, 152 families, 305 genera, and 449 species. The endophytic bacterial communities in *Tuber* ascomata were predominantly composed of Proteobacteria, accounting for an average of 99.50% of total sequences. Phyla with lower abundances included Firmicutes (0.12%), Actinobacteriota (0.08%), Bacteroidota (0.07%), and Acidobacteriota (0.03%), with all other phyla collectively comprising only 0.20% of total abundance.

At the genus level, the most abundant taxon was *Bradyrhizobium* (59.20%), followed by the top 10 relative abundance groups: *Allorhizobium–Neorhizobium–Para-rhizobium–Rhizobium* (ANPR) (10.80%), *Pseudomonas* (4.70%), *Polaromonas* (3.00%), *Mesorhizobium* (1.90%), *Variovorax* (1.90%), *Afipia* (1.40%), *Acidovorax* (1.20%), *Tardiphaga* (1.20%), and *Bosea* (1.10%).

Significant differences in endophytic bacterial abundances were observed across samples: *Bradyrhizobium* exhibited the highest content (99.80%) in *T. sinense* from

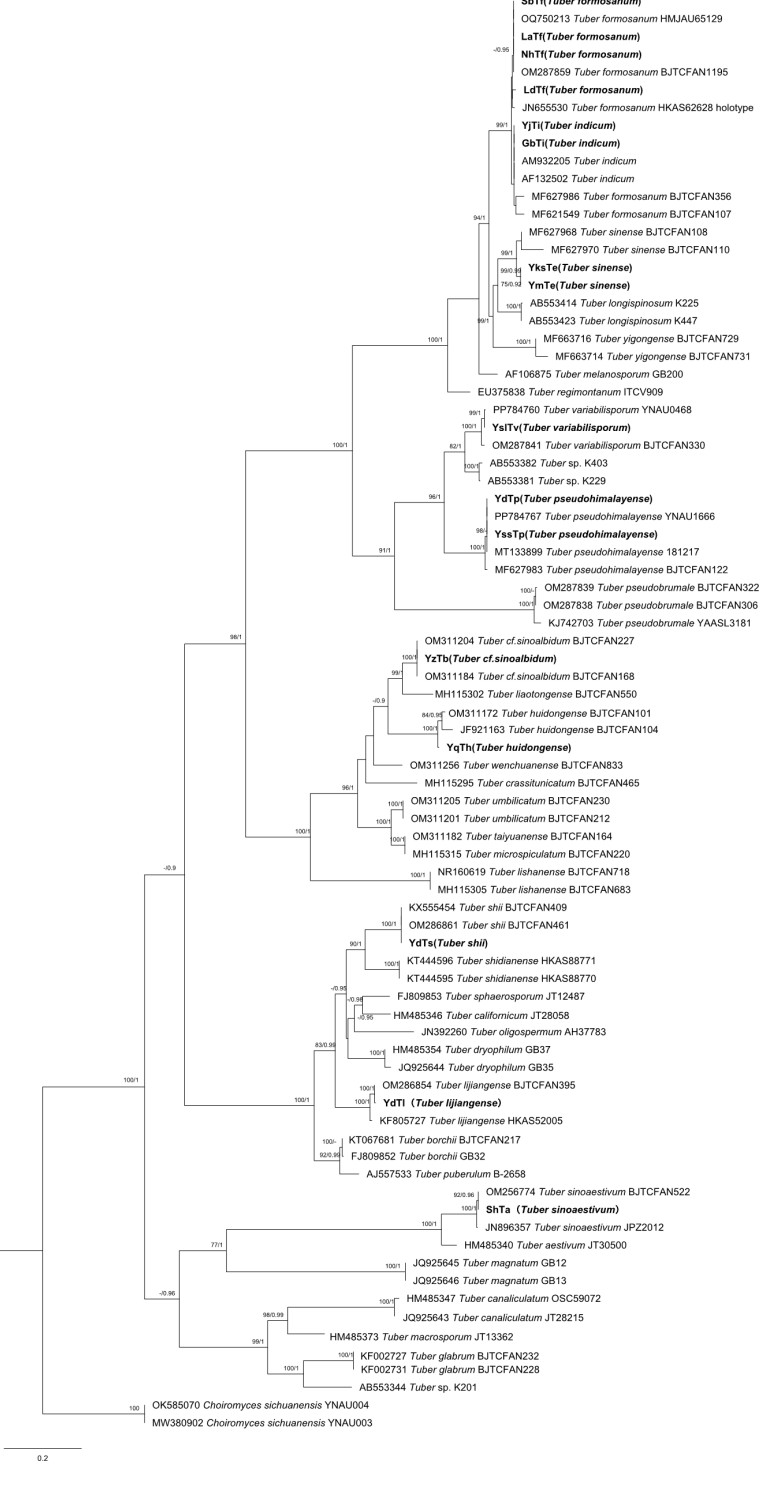

**FIG 1** Phylogenetic tree of the genus *Tuber* constructed using ITS data sets. The numbers at each branch node represent bootstrap support values derived from 1,000 replicates. Sequences produced in this contribution are shown in boldface type. Sequences of *C. sichuanensis* (accession numbers MW380902 and OK585070) were included as the outgroup in the analysis. Sample abbreviations follow those defined in Fig. 7.

Mengzi, Yunnan (YmTe), while the lowest content (7.90%) occurred in *T. shii* from Dali, Yunnan (YdTs). Conversely, ANPR dominated in the YdTs sample (55.40%) but was scarcely present (0.03%) in *T. formosanum* from Dalian, Liaoning (LdTf) (Fig. 2).

## Endophytic bacterial diversity and community structure variation

To investigate the effects of host species and geographic location on bacterial communities in ascomata, we conducted comparative analyses of community diversity based on the Shannon diversity index. Results revealed significant differences in bacterial diversity both across truffle species and geographic locations. Specifically, at the species level, *T. lijiangense* exhibited a relatively higher median Shannon diversity index (5.111) with greater data dispersion, whereas *T. sinense* had a lower median index (1.336) with more concentrated data distribution (Fig. 3A). A Kruskal–Wallis test further indicated statistically significant diversity differences among species ($P < 0.001$).

At the geographic distribution level, significant differentiation in endophytic bacterial community diversity was observed across sampling regions (Fig. 3B). The Dali, Yunnan (YD) group had the highest median Shannon index (4.5753) and the greatest data dispersion among all groups, while the Mengzi, Yunnan (YM) group displayed the lowest diversity level (median index = 0.5045) with the most concentrated data distribution. Further statistical analysis using Tukey's honest significant difference (HSD) test showed extremely significant diversity differences between the YM group and both the YD (Dali, Yunnan) and YQ (Qujing, Yunnan) groups, indicating that geographic location may conspicuously influence the assembly of endophytic bacterial communities.

β-Diversity was analyzed using non-metric multidimensional scaling (NMDS) and tested via Kruskal-Wallis tests (Fig. 4A and B). Additionally, principal coordinates analysis (PCoA) was applied to reduce dimensionality and visualize microbial community data from truffle samples (Fig. 4C). Analyses revealed distinct differences in bacterial communities among truffle fruiting bodies of different species and geographic origins. Among the investigated factors, host species and geographic location (Kruskal-Wallis test, sample size = 48) were identified as primary drivers of bacterial community composition.

## Core and key microorganisms

Following the characterization of bacterial community variations in *Tuber* ascomata, this study aimed to identify core microorganisms shared across different geographic conditions and truffle species. Within the same truffle species, fruiting bodies from distinct geographic locations harbored different numbers of unique endophytic bacteria. For example, *T. pseudohimalayense* exhibited wide fluctuations in the number of unique ASVs, ranging from 160 in the YssTp (Shaoshang, Kunming, Yunnan) sample to 1,327 in the YslTv (Shuanglong, Kunming, Yunnan) sample. Similarly, significant differences in

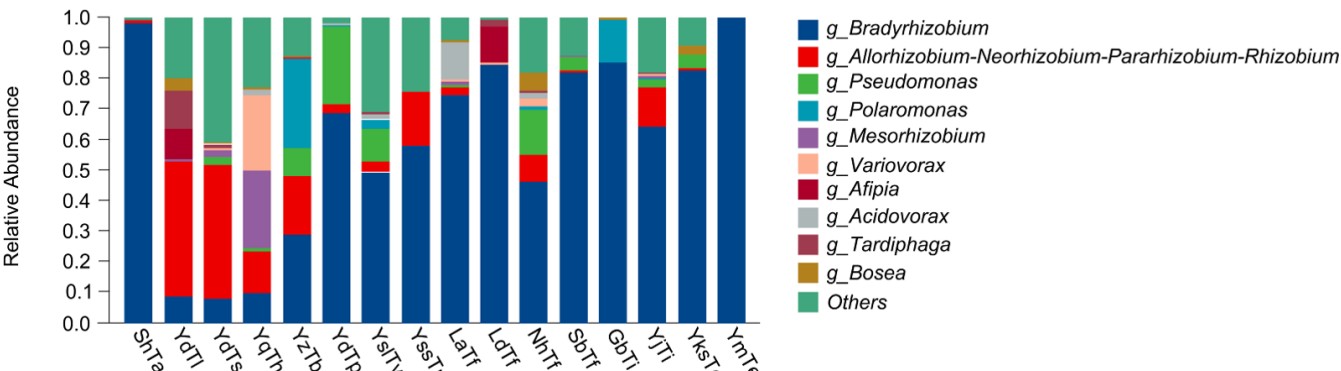

**FIG 2** Relative abundances of the top 10 endophytic bacterial genera in truffle fruiting bodies. Low-abundance and unclassified/unidentified taxa are grouped as "Others." Values represent the mean of three replicates. Sample abbreviations follow those defined in Fig. 7.

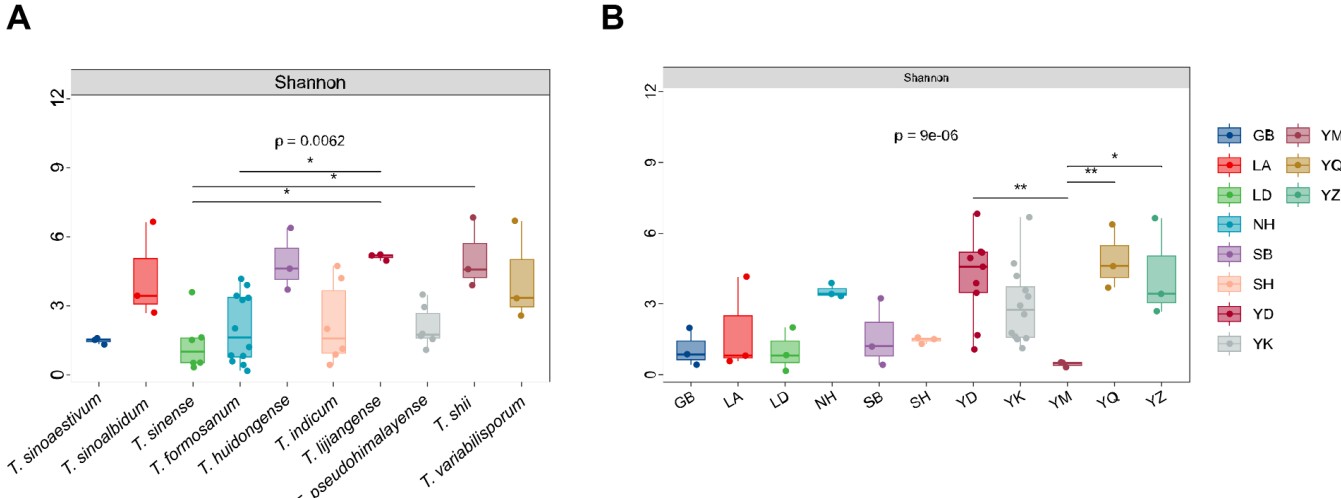

**FIG 3** Endophytic bacterial diversity in truffles influenced by host species (A) and geographic location (B). Horizontal lines above boxplots denote pairwise comparisons via *t*-tests, with asterisks indicating significance of mean differences (**$P < 0.01$; *$P < 0.05$). Dark horizontal lines indicate group pairs with significant differences (GB: Bijie, Guizhou; LA: Anshan, Liaoning; LD: Dalian, Liaoning; NH: Hohhot, Inner Mongolia; SB: Baoji, Shanxi; SH: Huidong, Sichuan; YD: Dali, Yunnan; YK: Kunming, Yunnan; YQ: Qujing, Yunnan; YZ: Zhaotong, Yunnan).

the number of unique endophytic bacteria were observed among different species at the same geographic location. In the Dali, Yunnan (Yd) region, *T. pseudohimalayense* (YdTp) contained 217 unique bacterial ASVs, whereas *T. shii* (YdTs) from the same region harbored as many as 1,222 unique ASVs (Fig. 5A). However, it is important to highlight that five bacterial ASVs were shared across all investigated truffle ascomata microbiomes, including three ASVs (ASV_1271, ASV_2731, ASV_3127) identified as *Bradyrhizobium* and two ASVs belonging to the ANPR complex (Fig. 5C).

LDA was used to quantify differences in microbial taxa between *T. lijiangense* from Dali, Yunnan (YdTl), and *T. huidongense* from Qujing, Yunnan (YqTh). Results showed that *Variovorax* exhibited a high LDA score approaching 5 in the YqTh group, indicating significant differential dominance and serving as a signature taxon for this group. In contrast, g_*Pseudoduganella*, f_D05-2, and g_D05-2 displayed moderate LDA scores in the YdTl group, suggesting relative differences and potential as characteristic taxa for the YdTl group. Notably, these scores were lower than that of *Variovorax* in the YqTh group, further highlighting the more pronounced differential significance of *Variovorax* in the YqTh group (Fig. 5B).

To further resolve the functional position of core microbes within the community interactome, we performed network topology analysis to delineate the modular architecture and role assignment of the bacterial assemblage (Fig. 6). The *Tuber*-associated endobacteria could be partitioned into nine functional modules (module_0 to module_8), each assigned to one of four topological roles on the basis of within-module connectivity ($Z_i$) and among-module connectivity ($P_i$): module hubs, network hubs, connectors, and peripherals. Module_3—dominated by *Bradyrhizobium*—was classified as peripheral (low $Z_i$ and $P_i$); nevertheless, because it contains core ASVs present in all samples, its high-abundance nitrogen-fixing capacity confers the status of a "functional core" on the community. Within module_1, which encompasses the ANPR complex, a small subset of connectors exhibited $P_i$ values of 0.4–0.6 and mediated cross-module coordination of nitrogen and phosphorus metabolism among module_1, module_3, and module_5 (containing *Pseudomonas*). Peripherals accounted for ≥85% of all nodes across modules; these low-abundance, environmentally responsive taxa (e.g., *Polaromonas*, *Bosea*) participate only in localized metabolic processes under specific geographic conditions.

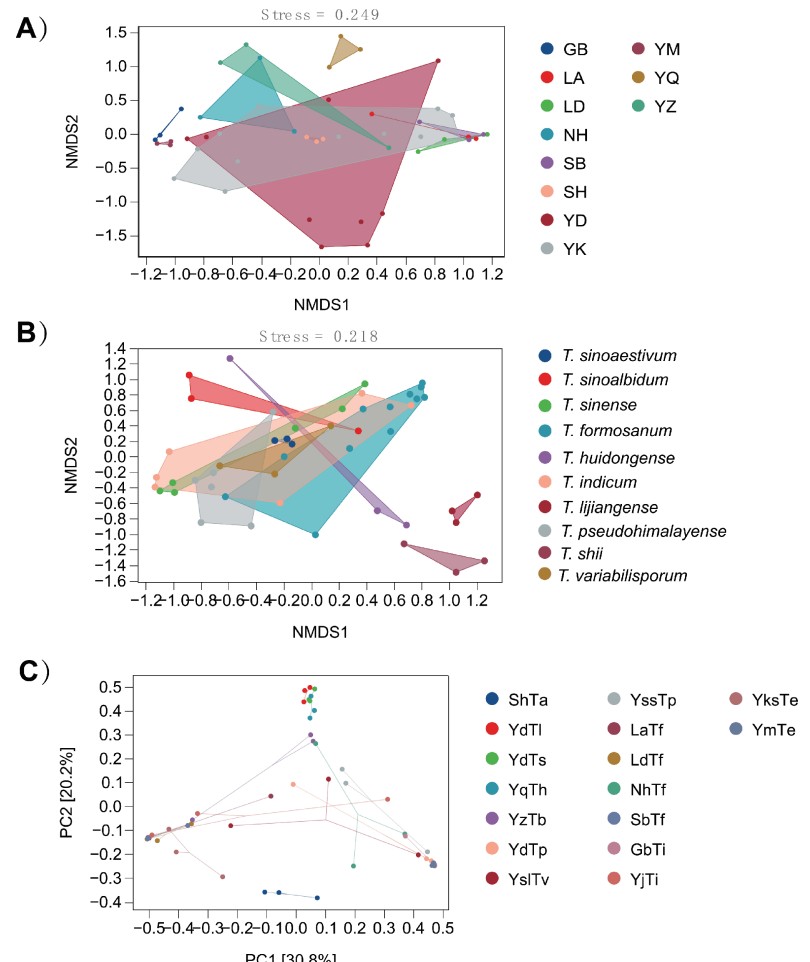

**FIG 4** Effects of geographic locations (A) and different host species (B) on the endophytic bacterial community structure of *Tuber* spp. Community structures are visualized via NMDS and PCoA based on Bray–Curtis dissimilarity matrices and (C). Sample abbreviations follow those defined in Fig. 3 and 7.

## DISCUSSION

### Abundance and composition of *Tuber* endophytic microorganisms

Our initial hypothesis (Hypothesis 1) proposed that conserved vertical transmission of bacterial microbiota would lead to convergence in endophytic communities across *Tuber* species, given their phylogenetic proximity. However, comparative analysis revealed statistically significant divergence in bacterial community structure among species. High-abundance taxa *Bradyrhizobium* and ANPR showed 8- to 10-fold variations between samples (e.g., YmTe vs YdTs).

The genus *Bradyrhizobium* consists of nitrogen-fixing bacteria that are ubiquitously distributed in both rhizosphere soils and root nodules of leguminous plants. Their multifunctional contributions to symbiotic systems encompass not only nitrogen assimilation but also phytohormone production and stress tolerance enhancement. These bacteria convert atmospheric nitrogen into available ammonia through efficient biological nitrogen fixation while assisting plants in absorbing essential minerals like phosphorus and potassium, thereby significantly improving host plant nutrition. *Bradyrhizobium* also secretes plant growth hormones (e.g., IAA) and signaling molecules that promote root development, directly influencing legume growth and development (31). Studies show their specific distribution in different legume tissues, primarily colonizing root nodules but also significantly present in rhizoplane and rhizosphere soils.

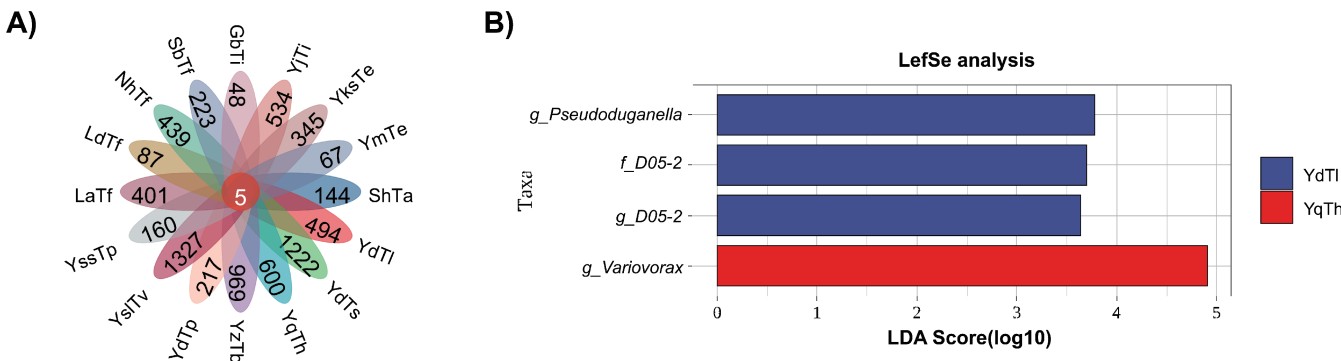

FIG 5 Core microbiome of *Tuber* samples. (A) Petal plots of core and unique bacterial amplicon sequence variants (ASVs), with petal labels indicating sample names and numbers reflecting ASV counts, highlighting compositional distribution and differences among samples (sample abbreviations follow those in Fig. 1). (B) Linear discriminant analysis effect size (LEfSe) plot, where the horizontal axis represents linear discriminant analysis (LDA) scores (log10 scale) measuring taxon effect sizes, and the vertical axis lists taxonomic groups. Blue bars: *T. lijiangense* from Dali, Yunnan (YdTl); red bars: *Tuber huidongense* from Qujing, Yunnan (YqTh), showing the relative importance of taxa in each sample group and identifying significantly differential biomarker taxa between the two groups. (C) Core ASVs in *Tuber* fruiting bodies, listing ASV numbers and taxonomic information at the family and genus levels to display shared bacterial taxa across samples. Sample abbreviations in panel A follow those defined in Fig. 7.

They help plants resist soil-borne pathogens by producing antimicrobial substances and inducing systemic resistance while regulating host gene expression to affect metabolic pathways and responses to abiotic stresses like drought and salinity (32).

In addition to *Bradyrhizobium*, *Tuber* ascomata were found to host substantial populations of ANPR and *Pseudomonas*. Symbiotic relationships between ANPR groups and mycorrhizal fungi have been widely confirmed (33), suggesting that these rhizobia may utilize truffle ascomata as new niches for ecological expansion, implying more diverse roles in mycorrhizal symbiotic systems than previously recognized. *Pseudomonas* can specifically recognize fungal-secreted signals like chitin and regulate colonization via quorum sensing. Their high abundance in truffle fruiting bodies may relate to specialized nutritional metabolism, particularly efficient utilization of fungal-derived organic nitrogen and phosphorus compounds (34, 35). Notably, ANPR and *Pseudomonas* community structures exhibit distinct dynamic changes during truffle maturation, suggesting key roles in ascomata development and functional maintenance (17).

Overall, potential explanations for significant differences in endophytic bacterial abundances across samples include (i) variations in ECM host plants; truffles associate with diverse hosts such as *Pinus armandii*, *Pinus yunnanensis*, and *Quercus mongolica*, which differ significantly across regions, thereby influencing endophytic bacterial abundances through ECM symbiotic relationships (36). (ii) After ECM establishment, physiological and biochemical processes in host plants are altered, changing root microenvironments and directly affecting endophytic bacterial growth. In terms of nutrient acquisition, ECM symbiosis enhances host plant uptake efficiency of key nutrients like nitrogen and phosphorus (37, 38), creating favorable conditions for endophytic bacteria adapted to specific nutrient environments and increasing their

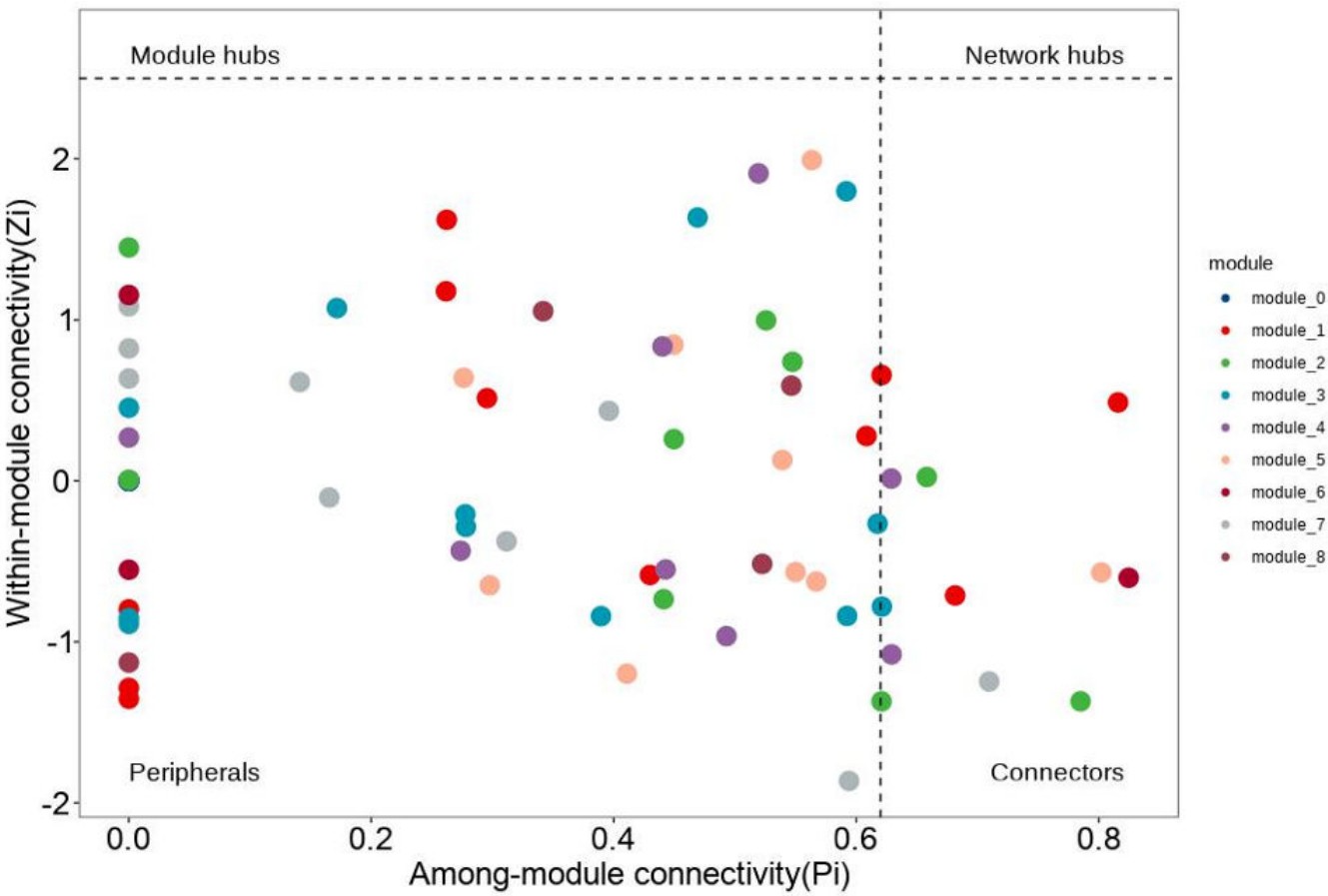

**FIG 6** Modular architecture and functional role assignment of the *Tuber*-associated endobacterial community. The *x*-axis indicates among-module connectivity (Pi), reflecting the strength of inter-module connections; the *y*-axis indicates within-module connectivity, reflecting the intensity of intra-module interactions. Dots of different colors represent the nine modules (module_0 to module_8). The labeled categories "Module hubs," "Network hubs," "Connectors," and "Peripherals" denote the four topological roles. The plot shows that peripherals constitute the majority of each module, whereas core functional taxa—exemplified by *Bradyrhizobium*-dominated module_3—maintain functional dominance through high abundance. Connectors serve as bridges that facilitate cross-module cooperation.

abundances, showing high populations of nitrogen-efficient bacteria in nitrogen-rich ECM microhabitats. ECM interactions also modify the composition and quantity of root exudates, differentially impacting bacterial populations: some exudates attract specific bacteria to promote proliferation, while others inhibit growth, altering bacterial community composition and abundance around roots (39). Additionally, ECM regulates plant immune responses and defense mechanisms, influencing endophytic bacterial communities—strengthened plant immunity via ECM produces antimicrobial substances that inhibit sensitive bacteria while selecting for bacteria with special adaptive mechanisms, thus shaping community structure and function (40, 41). These complex ECM–host interactions ultimately drive variations in endophytic bacterial abundance and dynamics.

## Key drivers influencing endophytic bacterial communities

The composition and diversity of endophytic bacterial communities in truffles are co-regulated by multiple biotic and abiotic factors. Through comprehensive analysis of microbiome data across distinct *Tuber* species and geographical regions, this study reveals the significant impacts of host species traits, geographical environmental disparities, and ECM symbiosis on truffle endophytic bacterial communities. Our findings are summarized as follows:

i. Host species-specific traits as core biotic drivers: interspecific biological differences within the genus *Tuber* are central to the differentiation of endophytic bacterial communities, and the reliability of host species identification is a prerequisite for elucidating this driving mechanism. Based on ITS sequence alignment and phylogenetic analysis, our 16 truffle samples were identified as 10 species (including *T. sinense*, *T. indicum*, *T. lijiangense*, among others), all of which formed stable species clusters with GenBank reference sequences (e.g., *T. indicum* AM932205, *T. sinense* MF627968) with bootstrap support values ≥ 90% (1,000 replicates). This precise taxonomic framework confirms that host species identity has greater explanatory power for bacterial community differentiation (PERMANOVA, $R^2 = 0.32$, $P < 0.001$) than geographical factors, laying a robust foundation for investigating species-specific drivers. These interspecific differences may be associated with species-specific metabolite profiles, prolonged fruiting body developmental cycles, and host interaction patterns (17, 42). For example, the extended developmental cycle of *T. lijiangense* (~6 months) provides a stable microenvironment for diverse bacterial colonization, whereas the rapid maturation of *T. sinoaestivum* (~3 months) may facilitate the dominance of specific taxa (e.g., *Bradyrhizobium* at 92.3%) via host filtering (40). Notably, the ITS phylogenetic tree revealed that *T. lijiangense* and *T. sinense* clustered in independent branches with 95% bootstrap support, and their bacterial Shannon diversity indices differed by 3.8-fold (5.111 vs 1.336, $P < 0.001$). Functional prediction further demonstrated that the abundance of "signal transduction" and "secondary metabolism" genes in *T. lijiangense* was 2.1 times higher than that in *T. sinense*, directly linking host developmental traits to bacterial community diversity. Additionally, interspecific variations in cell wall components (e.g., chitin content) likely modulate bacterial attachment through chemical signaling (43). *Pseudomonas* spp., which recognize fungal-derived chitin oligosaccharides and activate quorum sensing systems, exhibited elevated abundance in *T. shii* fruiting bodies (4.7% vs a 1.2% average in other species). Host genetic background also played a critical role: *Bradyrhizobium* dominated *T. sinense* from Mengzi, Yunnan (YmTe, 99.8%), but accounted for only 7.9% in *T. shii* from Dali, Yunnan (YdTs), representing a 12.6-fold difference. This genotype-driven selection was further corroborated by ITS-based species delimitation—for instance, *T. sinense* (YmTe, YksTe) from distinct regions consistently maintained high *Bradyrhizobium* abundance (>60%), whereas sympatric *T. pseudohimalayense* (YdTp, YssTp) harbored only 5%–12% of this genus, confirming the specificity of host species filtering. Furthermore, all 10 ITS-identified species harbored five core Rhizobiales ASVs (three *Bradyrhizobium*, two ANPR), collectively accounting for 61.1% of total bacterial abundance with fluctuations of less than 15% among conspecific samples from different regions. This suggested an evolutionarily conserved symbiotic relationship between Rhizobiales and the genus *Tuber*—notably, in the northeastern endemic *T. formosanum*, the core *Bradyrhizobium* ASV_3127 reached 48.7% abundance, potentially conferring nitrogen-fixing advantages for its survival in low-nitrogen soils (1.2–1.8 g kg$^{-1}$). This finding not only corroborates Le Roux et al.'s (10) conclusion regarding "obligate symbiosis between *Bradyrhizobium* and *Tuber*" but also extends the host range of this symbiosis by leveraging ITS-based species diversity data (44, 45).

ii. Geographical factors indirectly shape bacterial distribution: soil properties, climate, and host plant differences mediated by geography significantly influence endophytic bacterial communities. Shannon diversity differed sevenfold between Dali, Yunnan (YD, mean elevation ~3,000 m; 4.5753), and Mengzi, Yunnan (YM, ~1,500 m; 0.5045; Tukey HSD test). Dali samples exhibited higher diversity and dispersion, while Mengzi communities clustered tightly (Fig. 4B), likely linked to soil and microclimate disparities. Neutral pH (6.8 ± 0.3) and

high organic matter (25.7 ± 3.2 g/kg) in Dali soils promote oligotrophic bacteria (e.g., *Acidobacteriota*, 0.03%) and symbiotic nitrogen fixers (e.g., *Bradyrhizobium*, 59.2%). In contrast, acidic Mengzi soils (pH 5.2 ± 0.2) with low phosphorus (8.3 ± 1.5 mg/kg) select for acid-tolerant *Pseudomonas* (6.2%) (46). NMDS analysis revealed stronger geographical influence on beta-diversity (stress = 0.193) than species effects (stress = 0.249; Fig. 5B), with PERMANOVA confirming significant geographical impacts ($F = 3.14$, $P = 9.5 \times 10^{-15}$). Climatically, high rainfall regions (e.g., Qujing, Yunnan; annual precipitation 1,200 mm) enrich root-exuded sugars, favoring *Variovorax* (LDA = 4.82; Fig. 6B), which degrades aromatic compounds (47). Conversely, temperate monsoon climates (e.g., Dalian, Liaoning; mean annual temperature 8.8°C) limit Gram-negative bacteria, leaving *Bradyrhizobium* dominant (95.6%). These findings highlight how multidimensional niche differentiation driven by geography shapes region-specific bacterial communities.

iii. ECM symbiosis as a critical mediator: the ECM system links host plants, truffles, and endophytic bacteria via nutrient cycling and chemical signaling. ECM symbiosis enhances host nitrogen and phosphorus uptake (37), favoring bacteria with nutrient-use efficiency (e.g., *Rhizobiales*). *Bradyrhizobium* thrives by fixing atmospheric nitrogen and solubilizing soil phosphorus via organic acid secretion (48), dominating truffle microbiomes (mean abundance 59.2%). ECM-derived antimicrobial peptides (e.g., trufflecin in *T. shii*) inhibit 70% of non-symbiotic bacteria (49) but spare *Pseudomonas* with antioxidant enzymes (50), elevating its abundance in YdTs (12.3%). ECM-induced ROS bursts select oxidative stress-tolerant taxa (e.g., *Afipia*, 1.4%) via thiol metabolism (51).

The abovementioned key drivers of truffle endophytic bacterial communities do not act in isolation but synergistically shape microbial community structure and function through intricate interaction networks. The interactive effects of host species traits and geographical environmental factors significantly influence the stability of core microbiota. In Dali, Yunnan, *T. shii* (YdTs) samples exhibited a remarkably high relative abundance of ANPR taxa (55.40%; Fig. 3). This phenomenon may stem from the species' selective recruitment strategy in phosphorus-rich soil environments. The neutral pH (6.8 ± 0.3) and high organic matter content (25.7 ± 3.2 g/kg) of Dali soils provide an ideal habitat for ANPR strains with phosphate-solubilizing capabilities. Concurrently, *T. shii* likely prioritizes the recruitment of these phosphorus-cycling strains through species-specific metabolic traits and signaling mechanisms, resulting in ANPR dominance (52, 53). In stark contrast, ANPR abundance in *T. formosanum* (LdTf) samples from nutrient-poor soils in Dalian, Liaoning, was negligible (<0.03%). Environmental stresses in Dalian—including low temperatures (mean annual 8.8°C) under a temperate monsoon climate and soil nutrient scarcity—likely compromised host selection capacity, drastically reducing ANPR colonization. Furthermore, climate conditions may modulate the strength of ECM symbiosis. It has been demonstrated that drought stress reduces hyphal network density, intensifying dispersal limitations for bacterial communities (54). These interactions underscore the hierarchical nature of microbial community assembly. A comprehensive understanding of these microbial dynamics is essential for (1) predicting how truffle-associated endophytic bacterial communities will respond to climate change, and (2) developing science-based strategies to maintain ecosystem stability and preserve critical ecological functions.

By integrating functional genomic annotations with edaphic metadata, we propose a "impact–function–habitat coupling" framework. The *Bradyrhizobium–T. sinense* symbiosis coincides with severe nitrogen limitation at >3,000 m a.s.l.; the *nifH* gene repertoire of the enriched *Bradyrhizobium* OTUs ($n = 9$) predicts a 3.7-fold higher nitrogenase efficiency compared to low‐elevation conspecifics. Conversely, the ANPR consortium detected in *T. shii* harbors *gcd* (glucose dehydrogenase) and *phoD* alkaline phosphatase genes that mobilize organic phosphorus, an adaptation congruent with the high total P yet low available P of Dali soils. Thus, host species act as ecological filters,

amplifying microbial functions that compensate for specific habitat resource deficits, thereby transforming descriptive β-diversity patterns into testable eco-physiological mechanisms.

## Key microbial taxa

This study, employing high-throughput sequencing and multivariate statistical analyses, has identified some critical microbial taxa within truffle ascomata that likely play pivotal roles in ecological functionality and developmental processes. *Bradyrhizobium* emerged as a dominant genus across multiple samples, exhibiting a mean relative abundance of 59.20%, suggesting its role as a core member of truffle endophytic communities. Renowned for nitrogen-fixing capacity, *Bradyrhizobium* converts atmospheric nitrogen into plant-available ammonium, potentially providing essential nitrogen sources for truffles and their host plants. Additionally, its secretion of plant growth hormones (e.g., IAA) enhances root development and nutrient uptake, indirectly supporting truffle growth. Another key taxon, the ANPR group, displayed relative abundances exceeding 50% in some samples. While extensively studied in legume symbiosis, ANPR's prominence in truffles implies critical roles in nitrogen metabolism and nutrient cycling. Notably, ANPR abundance varied significantly across *Tuber* species and regions: it constituted >55% in *T. shii* (YdTs) from Dali, Yunnan, likely linked to its *PhoC* gene (encoding acid phosphatase), which hydrolyzes organic phosphorus compounds in high-organic-matter soils (52, 53). Conversely, ANPR abundance plummeted in phosphorus-poor environments, such as *T. formosanum* (SbTf) from Baoji, Shaanxi, underscoring its regulation by environmental resource availability. *Pseudomonas* (with a mean relative abundance of 4.70%) represents another functionally versatile taxon, producing secondary metabolites (e.g., antibiotics and volatile organic compounds) that may contribute to truffle aroma biosynthesis or pathogen defense (55). Its ability for organic phosphorus mineralization further enhances the adaptability of truffles to survive in phosphorus-limited soils (56).

LEfSe (LDA threshold > 4.0) identified *Variovorax* as a highly discriminatory taxon in *T. huidongense* (YqTh) from Qujing, Yunnan (LDA = 4.91; abundance 3.2%, 6.4× higher than other groups). In contrast, *Pseudoduganella* (LDA = 3.82) and family *f_D05-2* (LDA = 3.65) were enriched in *T. lijiangense* (YdTl) from Dali, Yunnan (Fig. 6B). These taxa likely contribute to species-specific ecological functions. Consistent with our hypothesis, all *Tuber* species harbored five core endophytic bacterial taxa belonging to the order *Rhizobiales* (Fig. 6C). The ubiquity and ecological dominance of Rhizobiales imply a functionally indispensable role for the truffle species studied. Studies highlight their nitrogen-fixing proficiency (57), particularly in symbiotic relationships with legumes, where they enhance soil nitrogen availability and reduce reliance on synthetic fertilizers (58, 59). Additionally, certain *Rhizobiales* members optimize nutrient acquisition and stress resilience in plants (60, 61). Their functional impacts vary across species and ecosystems, warranting further exploration of their ecological versatility (62, 63).

## Conclusions

This study systematically analyzed the endophytic microbiota of truffle (*Tuber* spp.) ascomata across diverse geographical regions and species in China, revealing key ecological distribution patterns of their bacterial communities. Results demonstrate a pronounced dominance of *Bradyrhizobium* within truffle endophytic bacterial assemblages, with its relative abundance significantly surpassing other genera, suggesting its paramount ecological role in truffle–microbe interaction networks. Furthermore, this study demonstrates that geographical locations and species identity critically shape bacterial diversity and community structure. Distinct *Tuber* species create unique ecological niches that modulate bacterial composition and abundance. Shared core bacterial taxa across *Tuber* species, particularly members of the order *Rhizobiales*, likely contribute to nitrogen fixation and plant growth promotion. Understanding the dynamics of the community structure of endophytic bacterial communities within

truffle ascomata represents a critical research focus for advancing microbial ecology and elucidating complex fungal–bacterial symbiotic relationships. Collectively, these findings establish a scientific foundation that provides critical insights that might contribute to sustainable truffle resource management, optimized cultivation techniques, and effective conservation strategies for wild populations. This research thereby also contributes in these ways to the promotion of sustainable development of truffle-related industries.

## MATERIALS AND METHODS

### Sample collection and processing

Truffle ascomata were collected from 16 geographically distinct sites across six provinces between October and December 2024 (Fig. 7). At each sampling location, five mature ascomata were rigorously selected, excluding immature or rotting individuals to ensure consistency and reliability in sample quality. Freshly collected truffles were immediately placed in sterile, sealed bags at the collection site and transported to the laboratory under 4°C controlled cold storage conditions to ensure optimal preservation of their original physiological state. Upon arrival to the laboratory, ascomata were assigned a unique identifier to maintain traceability and facilitate standardized sample processing throughout the study.

Subsequently, in the laboratory, the ascomata surfaces were carefully rinsed with sterile water to remove adhering soil and debris. Washed ascomata were air-dried in a laminar flow hood; once surface-dried, their surfaces were sterilized by wiping with 75% ethanol to eliminate potential microbial contamination. Following surface sterilization, ascomata were aseptically dissected by manually separating the internal gleba from the outer peridium using a sterilized scalpel, ensuring exclusive sampling of the inner truffle tissue (4). Flesh tissues from the five ascomata making up each sample were pooled, homogenized, and equally distributed into sterilized 5-mL centrifuge tubes. To ensure data reliability and accuracy, three biological replicates were prepared for each sample. Processed samples were immediately stored at −80°C until use in subsequent microbial community analyses.

### ITS sequencing for species identification of *Tuber* samples

DNA was extracted from newly collected samples using the CTAB (cetyltrimethylammonium bromide) method (64). The ITS region was amplified with the universal primer pair ITS1/ITS4 (Qingke Biotech, Kunming) (65). The 30 µL polymerase chain reaction (PCR) system contained 15 µL of 2 × PCR mix, 0.75 µL of each forward and reverse primer (5 µM), 2 µL of template DNA, and 11.5 µL of deionized water. The PCR protocol was as follows: initial denaturation at 95°C for 3 min, followed by 33 cycles of 95°C for 30 s (denaturation), 52°C for 1 min (annealing), and 72°C for 8 min (extension). PCR products were subsequently sequenced by Qingke Biotech (Beijing) using the same primers.

The total length of the amplified ITS sequences was approximately 650 bp. After trimming low-quality regions at both ends, approximately 600 bp of high-quality sequences was retained for subsequent phylogenetic analysis. These sequences were aligned against the GenBank database using BLAST (2.13.0) and further processed with MEGA11.0 software. A phylogenetic tree was constructed using the neighbor-joining (NJ) method (66), with statistical support for the NJ tree evaluated via non-parametric bootstrap resampling (1,000 replicates). Through phylogenetic comparison and tree construction, the 16 collected samples were classified into 10 distinct species. All sequences have been deposited in GenBank under accession numbers PV664867–PV664882.

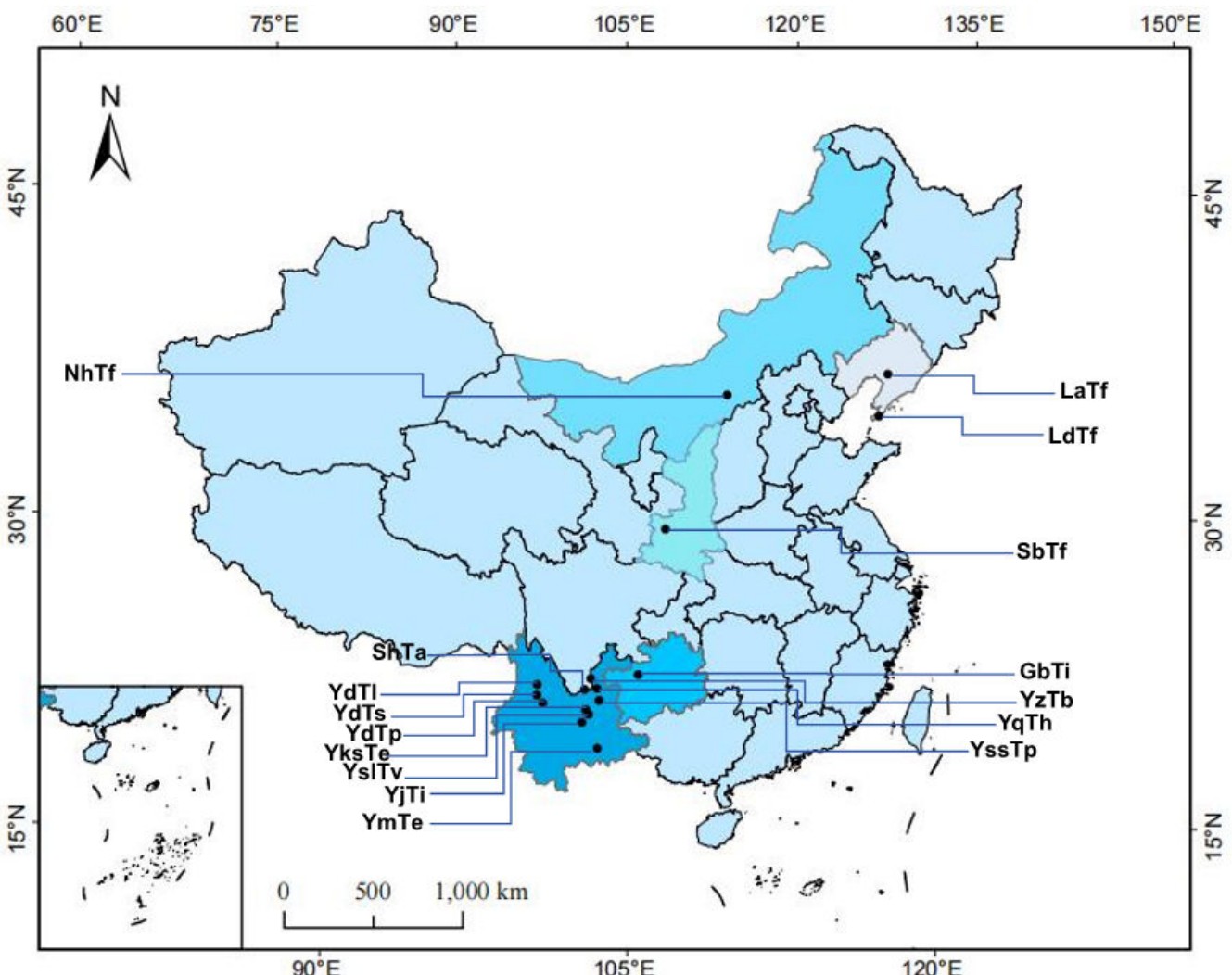

**FIG 7** Distribution map of truffle fruiting body samples collected in this study (YmTe: Mengzi City, Honghe Hani and Yi Autonomous Prefecture, Yunnan Province [*T. sinense*]; YjTi: Jinning District, Kunming City, Yunnan Province [*T. indicum*]; YslTv: Shuanglong Subdistrict, Panlong District, Kunming City, Yunnan Province [*T. variabilisporum*]; YksTe: Songhuaba, Dianyuan Subdistrict, Panlong District, Kunming City, Yunnan Province [*T. sinense*]; YdTp: Dali Bai Autonomous Prefecture, Yunnan Province [*T. pseudohimalayense*]; YdTs: Dali Bai Autonomous Prefecture, Yunnan Province [*T. shii*]; YdTl: Dali Bai Autonomous Prefecture, Yunnan Province [*T. lijiangense*]; ShTa: Bailadu Village, Canyuhe Town, Huidong County, Sichuan Province [*Tuber sinoaestivum*]; YssTp: Shaoshang Village, Panlong District, Kunming City, Yunnan Province [*T. pseudohimalayense*]; YqTh: Huize County, Qujing City, Yunnan Province [*T. huidongense*]; YzTb: Qiaojia County, Zhaotong City, Yunnan Province [*Tuber cf. sinoalbidum*]; GbTi: Dafang County, Bijie City, Guizhou Province [*T. indicum*]; SbTf: Chencang District, Baoji City, Shaanxi Province [*Tuber formosanum*]; LdTf: Dalian City, Liaoning Province [*T. formosanum*]; LaTf: Haicheng City, Anshan City, Liaoning Province [*T. formosanum*]; NhTf: Hohhot, Inner Mongolia Autonomous Region [*T. formosanum*]).

## 16S rRNA sequencing for bacterial identification from *Tuber* samples

Samples were removed from the −80℃ freezer, and appropriate amounts (0.2–0.5 g) were quickly transferred into centrifuge tubes containing extraction lysis buffer for grinding. Grinding was performed using a Tissuelyser-48 multi-sample tissue grinder (Shanghai Jingxin Company, China) at a frequency of 60 Hz. Genomic DNA was extracted using the MagBeads FastDNA Kit for Soil (116564384; MP Biomedicals, CA, USA) following the manufacturer's protocol. The V5–V7 hypervariable regions of the bacterial 16S rRNA gene were amplified using the primer pair: forward primer F: AACMGGATTAGATACCCKG and reverse primer R: ACGTCATCCCCACCTTCC (67). After PCR amplification, bacterial

amplicons were sequenced on an Illumina MiSeq PE250 platform (Personalbio, Shanghai, China) using paired-end sequencing.

Raw bacterial reads were processed using the QIIME 2 pipeline. Initial steps included quality trimming, barcode sorting, and removal of chimeric and singleton sequences. Filtered reads were clustered into ASVs with ≥ 99% similarity using the taxonomically annotated 16S rRNA Greengenes database (68). The raw sequence data have been deposited in the NCBI Sequence Read Archive under accession numbers PRJNA1227731 and PRJNA1243022.

## Statistical analysis

One-way analysis of variance followed by Tukey HSD tests was employed to compare differences in bacterial abundance and diversity (Shannon index) among *Tuber* fruiting bodies from distinct geographic locations. PCoA and NMDS were used to visually characterize similarities and dissimilarities in microbial community structures across samples (69). PERMANOVA was performed to assess significant differences in microbial community structures among *Tuber* ascomata bodies grouped by geographic origin and truffle species.

Core and unique microbial taxa were identified from the endophytic ASV table and visualized using petal plots. LEfSe was applied to detect microbial groups with significant differential abundances across groups at multiple taxonomic levels, with their contributions to group differentiation evaluated via LDA scores.

## ACKNOWLEDGMENTS

The authors acknowledge the technical infrastructure support from the Kunming Institute of Botany, Chinese Academy of Sciences, particularly their professional assistance in high-throughput sequencing and data analysis. Special appreciation is extended to Shanghai Personalbio Co., Ltd., for their expert services in Illumina MiSeq sequencing technology, which ensured high-quality data acquisition for microbial community analysis. Jesús Pérez-Moreno acknowledges the support from Colegio de Postgraduados for an academic stay in China. Furthermore, they gratefully acknowledge the constructive feedback from anonymous peer reviewers, whose insightful comments significantly enhanced the scientific rigor and academic quality of this study.

They express their sincere gratitude for the financial support provided by the National Key Research and Development Program (grant no. 2024YFF1306703), the Project of Guangxi Institute of Industrial Technology Research (CYY-HT2023-JSJJ-0038), the Designated Support Project of Chinese Academy of Sciences (KCXFZJ-DDBF-202403), the Yunnan Revitalization Talent Support Program to Jesús Pérez-Moreno, and the Yunnan Technology Innovation Program (202205AD160036) to Fuqiang Yu.

## AUTHOR AFFILIATIONS

[1]College of Horticulture, Hunan Agricultural University, Changsha, Hunan, China
[2]Yunnan Key Laboratory for Fungal Diversity and Green Development & Yunnan International Joint Laboratory of Fungal Sustainable Utilization in South and Southeast Asia, Germplasm Bank of Wild Species, Kunming Institute of Botany, Chinese Academy of Sciences, Kunming, Yunnan, China
[3]Institute of Subtropical Agriculture, Chinese Academy of Sciences, Hunan, China
[4]Huanjiang Agriculture Ecosystem Observation and Research Station of Guangxi, Guangxi Key Laboratory of Karst Ecological Processes and Services, Huanjiang Observation and Research Station for Karst Ecosystems, Chinese Academy of Sciences, Huanjiang, China
[5]College of Resources and Environment, Yunnan Agricultural University, Kunming, Yunnan, China
[6]Colegio de Postgraduados, Campus Montecillo, Edafología, México

## AUTHOR ORCIDs

Xunyang He ⓘ http://orcid.org/0000-0002-0536-786X
Dong Liu ⓘ http://orcid.org/0000-0002-4271-0591

## AUTHOR CONTRIBUTIONS

Man Guo, Data curation, Formal analysis, Software, Writing – original draft | Zhilan Xia, Project administration, Supervision | Xunyang He, Funding acquisition, Project administration, Resources | Shanping Wan, Software, Supervision | Yanliang Wang, Writing – review and editing | Shaolin Fan, Supervision | Jesús Pérez-Moreno, Writing – review and editing | Zhenyan Yang, Supervision | Chengmo Yang, Supervision | Dong Liu, Data curation, Writing – review and editing | Fuqiang Yu, Funding acquisition, Methodology, Project administration, Resources, Writing – review and editing

## DATA AVAILABILITY

All sequences have been deposited in GenBank under the accession numbers PV664867–PV664882. The raw sequence data have been deposited in the NCBI Sequence Read Archive under accession numbers PRJNA1227731 and PRJNA1243022.

## ADDITIONAL FILES

The following material is available online.

### Open Peer Review

**PEER REVIEW HISTORY (review-history.pdf).** An accounting of the reviewer comments and feedback.

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
