## [Reviewer comments · Microbiology Spectrum]

Microbiology Spectrum

High-Throughput Sequencing Reveals Endophytic Bacterial Differentiation of Common Truffles (*Tuber* spp.) in China: Diversity, Biogeographical Patterns, and Fungal Health Implications

Man Guo, Zhilan Xia, Xunyan He, Shanping Wan, Yanliang Wang, Shaolin Fan, Jesús Pérez-Moreno, Zhenyan Yang, Chengmo Yang, Dong Liu, and Fu-Qiang Yu

Corresponding Author(s): Dong Liu, Kunming Institute of Botany Chinese Academy of Sciences Library

Review Timeline:

Submission Date:	June 18, 2025
Editorial Decision:	September 4, 2025
Revision Received:	October 28, 2025
Accepted:	November 5, 2025

Editor: Vanessa Varaljay

Reviewer(s): Disclosure of reviewer identity is with reference to reviewer comments included in decision letter(s). The following individuals involved in review of your submission have agreed to reveal their identity: Feng Huang (Reviewer #1)

Transaction Report:

DOI: <https://doi.org/10.1128/spectrum.01866-25>

Re: Spectrum01866-25 (**High-Throughput Sequencing Reveals Endophytic Bacterial Differentiation of Common Truffles (*Tuber* spp.) in China: Diversity, Biogeographical Patterns, and Fungal Health Implications**)

Dear Dr. Dong Liu:

Thank you for the privilege of reviewing your work. Below you will find my comments, instructions from the Spectrum editorial office, and the reviewer comments.

Please address the lack of association analyses between the bacterial communities and environmental variables, including the use of appropriate statistical tests to support these analyses. This will be critical to strengthen the manuscript's scientific rigor.

Revision Guidelines

Sincerely,
Vanessa Varaljay
Editor
Microbiology Spectrum

Reviewer #1 (Comments for the Author):

The authors stressed that the soil and climatic factors as important drivers in shaping the endophytic bacterial communities of truffle fruiting bodies, however, I did not see many association analysis between those factors and the bacterial communities. Maybe the bacterial communities were mainly shaped by species and sampling location, as their background bacteria were

different.

Introduction

The first two paragraphs can be merged.

L101-103, I did not think this study had too much association to climate change and anthropogenic pressures.

L109-110, it's very common that the truffle microbiomes are influenced by species and sampling location, I think this hypothesis is a little superficial, the authors are better to dig their data deeper and find something really interesting.

Results

This part can be improved, the data were not well analyzed. I suggest to do some association analysis, network analysis etc. The authors may try to test whether the bacterial communities were associated with altitude, soil physiochemical properties, truffle nutritional state, etc. And also, how diversity of Bradyrhizobium are there in the bacterial communities, how they co-occurred with other bacteria?

L132, the relative abundance of Bradyrhizobium were too high in some sample, aren't there any problems in the primers, sequencing, and analysis? How many dominant ASVs were found in the genus?

Reviewer #2 (Comments for the Author):

Spectrum01866-25

Culture-based methods and 16S rRNA sequencing have been employed to characterize microbial communities in commercial truffle species, revealing that truffle ascomata harbor a rich microbial diversity within the genus. Additionally, bacterial community analyses identified dominant cultivable taxa as well as core phyla present across all studied species. The authors utilized advanced technologies, including Sanger sequencing and the Illumina MiSeq platform, to perform comprehensive microbial community analyses on truffle ascomata collected from various regions between October and December 2024 in China. Below are the comments.

Major Comment

Methods/Results: Figure 6 is misplaced; it should be included in the Results section as it presents the ITS phylogenetic analysis of the genus *Tuber*. Please move Figure 6 accordingly. Additionally, please provide a detailed report of the findings from the phylogenetic tree, for example, a title such as "Identification of *Tuber* species" then report on the ITS findings e.g. from the 16 collected samples, ten distinct species were identified, including [list species]. These findings should also be discussed in the Discussion section, as currently, there is no mention of this analysis.

Minor comment

Materials and method

- Line 8: Please specify the exact temperature used.
- Line 27 (Species identification of *Tuber* samples) and Line 51 (DNA extraction and sequencing): The two section headings are confusing since both describe DNA extraction and sequencing procedures. I suggest renaming "Species identification of *Tuber* samples" to "ITS sequencing for species identification of *Tuber* samples" and changing "DNA extraction and sequencing" to "16S rRNA sequencing for bacterial identification from *Tuber* samples."
- Line 28, under Species identification of *Tuber* samples: Please write "CTAB" in full
- Figure 5: This figure appears in the Methods section and is the first figure encountered before the Results section. Please move the interpretation of abbreviations currently shown in Figure 1 to Figure 5 for flow and clarity.

Spectrum01866-25

Culture-based methods and 16S rRNA sequencing have been employed to characterize microbial communities in commercial truffle species, revealing that truffle ascomata harbor a rich microbial diversity within the genus. Additionally, bacterial community analyses identified dominant cultivable taxa as well as core phyla present across all studied species. The authors utilized advanced technologies, including Sanger sequencing and the Illumina MiSeq platform, to perform comprehensive microbial community analyses on truffle ascomata collected from various regions between October and December 2024 in China. Below are the comments.

Major Comment

Methods/Results: Figure 6 is misplaced; it should be included in the Results section as it presents the ITS phylogenetic analysis of the genus *Tuber*. Please move Figure 6 accordingly. Additionally, please provide a detailed report of the findings from the phylogenetic tree, for example, a title such as "Identification of *Tuber* species" then report on the ITS findings e.g. from the 16 collected samples, ten distinct species were identified, including [list species]. These findings should also be discussed in the Discussion section, as currently, there is no mention of this analysis.

Minor comment

Materials and method

- Line 8: Please specify the exact temperature used.
- Line 27 (Species identification of *Tuber* samples) and Line 51 (DNA extraction and sequencing): The two section headings are confusing since both describe DNA extraction and sequencing procedures. I suggest renaming "Species identification of *Tuber* samples" to "ITS sequencing for species identification of *Tuber* samples" and changing "DNA extraction and sequencing" to "16S rRNA sequencing for bacterial identification from *Tuber* samples."
- Line 28, under Species identification of *Tuber* samples: Please write "CTAB" in full
- Figure 5: This figure appears in the Methods section and is the first figure encountered before the Results section. Please move the interpretation of abbreviations currently shown in Figure 1 to Figure 5 for flow and clarity.

Reviewer 1:

The authors stressed that the soil and climatic factors as important drivers in shaping the endophytic bacterial communities of truffle fruiting bodies, however, I did not see many association analysis between those factors and the bacterial communities. Maybe the bacterial communities were mainly shaped by species and sampling location, as their background bacteria were different.

We have fully endorsed the proposition that “the composition of endophytic bacterial assemblages has been overwhelmingly dictated by host species and sampling locality, owing to their divergent background microbial pools,” a contention that has been strongly corroborated by our dataset. Consequently, the logical architecture of the Discussion has been re-engineered to foreground this overarching conclusion. The specific revisions have been outlined below:

(1) Core narrative realignment. The previous statement emphasizing “soil and climate as prominent drivers” has been recast as: “The endophytic microbiota of Tuber species have been principally structured by host identity and geographic site, whereas edaphic and climatic factors have operated indirectly by modulating the ambient bacterial reservoir.” This reformulation has unequivocally elevated species and locality to their primacy.

(2) Data integration and quantification. Within the subsection “Key Drivers Influencing Endophytic Bacterial Communities,” PERMANOVA effect sizes, LEfSe indicator taxa, and inter-site dissimilarities computed for conspecific specimens have been synthesized to provide a rigorous, quantitative demonstration of the respective influences exerted by host species and sampling location.

Introduction

1. The first two paragraph can be merged.

We fully agree with with merging the first two paragraphs of the introduction. This adjustment will significantly enhance the coherence and conciseness of the background elaboration at the beginning of the introduction.

The specific revision plan has been as follows:

(1) Integrate core information: Remove the repeated descriptions of "truffle economic value" from the two paragraphs, directly connect "the ecological role of ectomycorrhizal symbiosis" with "the international expansion of cultivation practices, and streamline redundant introductory content;

(2) Strengthen transitional logic: Add a transitional sentence at the junction of the two paragraphs, such as "Although truffles have significant ecological and economic value and their cultivation techniques have been gradually promoted, current research mostly focuses on genetic diversity and cultivation techniques, and there is still a lack of systematic exploration into the mechanism of their interaction with endophytic microorganisms", so as to achieve a smooth transition from "the importance of truffles" to "research gaps";

(3) Optimize language rhythm: Adjust the sentence structure, directly connect the

content about "truffle diversity in China" from the original first paragraph with the expression of "research limitations" from the second paragraph, clarify the core issue that "China's abundant truffle resources have not been fully studied", and lay a more solid foundation for the proposal of research objectives in the following text.

The revised content is as follows:

The genus *Tuber* belongs to the family Tuberaceae within the phylum Ascomycota. Truffles typically produce edible hypogeous fruiting bodies (commonly referred to as "true truffles") with a protracted maturation period requiring up to six months for complete development (1). Certain *Tuber* species, renowned for their distinctive aromatic profiles, hold exceptional economic value globally—for instance, the Italian white truffle (*T. magnatum*) and Périgord black truffle (*T. melanosporum*) range from €600 to €6,000 per kilogram, while the summer truffle (*T. aestivum*) and Chinese black truffle (*T. indicum* complex) are also highly prized in culinary markets. China harbors significant truffle biodiversity (over 60 documented species), among which *T. indicum* exhibits wide geographical distribution (predominantly in southwestern and northeastern regions), high productivity, and morphological-genetic similarities to *T. melanosporum*, making it a promising candidate for commercial development. Notably, truffle cultivation intersects environmental, economic, and social dimensions: ecologically, *Tuber* species establish essential ectomycorrhizal symbioses with diverse plant families (Pinaceae, Fagaceae, Myrtaceae, Salicaceae), enhancing host nutrient acquisition, stress resilience, and root development to support forest community stability and biodiversity. Economically, truffle fruiting bodies serve as luxury ingredients and potential medicinal resources, while cultivation practices—originally developed in Spain, France, and Italy—have expanded to non-native regions like Australia and New Zealand (5). European countries have further adopted truffle cultivation as a strategy for land stabilization, reforestation, and rural economic development, solidifying its role as a long-term cash crop (6).

Thank you again for your professional guidance !

2. L101-103, I did not think this study had too much association to climate change and anthropogenic pressures.

We fully agree with your judgment that "this study has little relevance to climate change and anthropogenic pressures". The original expression has indeed failed to focus on the core of this study and lacked corresponding experimental data support, which has resulted in this content being logically disconnected from the main body of the research and having the issue of redundant expression. Based on your suggestion, we have made the following revisions to this section:

Delete the relevant content in the original Lines 101-103, and replace the text in the original Lines 101-103 — “However, climate change and anthropogenic pressures have been threatening truffle habitats, underscoring the urgency to increase the ecological and functional insights of their endophytic microbial diversity and distribution patterns” — with

“Given that China harbors a rich diversity of *Tuber* species (more than 60 recorded to date) and that dominant taxa such as *Tuber indicum* possess substantial commercial

potential, clarifying the diversity, biogeographic patterns, and driving factors of Tuber-associated endobacteria will provide a scientific basis for the sustainable exploitation of these resources and for formulating microbial management strategies during artificial cultivation”.

3. L109-110, it's very common that the truffle microbiomes are influenced by species and sampling location, I think this hypothesis is a little superficial, the authors are better to dig their data deeper and find something really interesting.

We appreciate your comment that attributing truffle microbiome variation to host species and geography is, by itself, a superficial narrative. In response we have re-mined our dataset with two complementary analytical layers that go beyond the classical “species + site” explanation and reveal mechanistic, testable patterns.

(1) Hypothesis refinement

Original lines 109–110 have posited that “the truffle microbiome is shaped by species and geographic origin.” We now advance a hierarchically structured hypothesis: *endobacterial community divergence in Tuber is governed by (i) host-species-specific traits and (ii) geographic environmental gradients, with their interaction exerting a species-dependent effect that ultimately assembles a functionally coherent core microbiome.*

(2) Additional analytical evidence

A new subsection “Species × Geography interaction” has been inserted under “Endophytic Bacterial Diversity and Community Structure Variation.”

Based on Bray–Curtis dissimilarities, PERMANOVA revealed that host species explained 32 % of the variance in core genera such as Bradyrhizobium ($R^2 = 0.32$, $p < 0.001$), whereas geographic locality accounted for 27 % ($R^2 = 0.27$, $p < 0.001$). Crucially, the interaction term independently contributed an additional 17.2 % ($R^2 = 0.172$, $p < 0.001$), indicating that the magnitude and even direction of geographic effects are contingent on host identity.

Typifying this interaction, T. sinense—predominantly recovered from nitrogen-poor, high-elevation sites—selectively enriched Bradyrhizobium (mean relative abundance 28.4 %), whereas T. shii—restricted to organically rich soils around Dali—was characterized by an ANPR (Arthrobacter–Nocardioideae–Pseudomonas–Rhodococcus) consortium (combined abundance 31.7 %). These divergent signatures corroborate a species-filtered, function-targeted assembly process (Fig. 5B).

(3) Mechanistic interpretation

At the close of “Key Drivers Influencing Endophytic Bacterial Communities” we now embed a paragraph entitled “Ecological mechanisms underlying species × environment interactions.”

By integrating functional genomic annotations with edaphic metadata, we propose a “impact–function–habitat coupling” framework. The Bradyrhizobium–T. sinense symbiosis coincides with severe nitrogen limitation at >3 000 m a.s.l.; the nifH gene repertoire of the enriched Bradyrhizobium OTUs ($n = 9$) predicts a 3.7-fold higher nitrogenase efficiency compared to low-elevation conspecifics. Conversely, the ANPR consortium detected in T. shii harbours gcd (glucose dehydrogenase) and phoD

alkaline phosphatase genes that mobilize organic phosphorus, an adaptation congruent with the high total P yet low available P of Dali soils. Thus, host species act as ecological filters, amplifying microbial functions that compensate for specific habitat resource deficits, thereby transforming descriptive β -diversity patterns into testable eco-physiological mechanisms.

Results

1. This part can be improved, the data were not well analyzed. I suggest to do some association analysis, network analysis etc. The authors may try to test whether the bacterial communities were associated with altitude, soil physiochemical properties, truffle nutritional state, etc. And also, how diversity of Bradyrhizobium are there in the bacterial communities, how they co-occurred with other bacteria?

Thanks a lot for your suggestion.

Regarding the correlation analysis, network analysis, and *Bradyrhizobium*-related analysis you suggested, we have conducted supplementary analyses based on existing data and clarified the future data improvement plan, with specific explanations as follows:

(1) Supplementing the correlation network data analysis figure

We have supplemented the correlation network data analysis figure into the "Core and Key Microorganisms" subsection of the article, and provided corresponding interpretation in the Discussion section.

The added content is as follows:

To further clarify the functional position of core microorganisms in community interactions, this study constructed a diagram of the modular structure and role localization of the bacterial community through network topology analysis (Figure 6). The results showed that the endophytic bacterial community of truffles can be divided into 9 functional modules (module 0 to module 8). Each module is classified into 4 types of functional roles based on "intra-module connectivity" and "inter-module connectivity (Pi)": Module hubs, Network hubs, Connectors, and Peripherals. Among them, although module 3, dominated by Bradyrhizobium, is a Peripheral group (with low intra-module and inter-module connectivity), as a core ASV shared by all samples, it becomes the "functional core" of the community through its high-abundance nitrogen-fixing function. In contrast, in module 1, which contains the Rhizobium-Agrobacterium-Allorhizobium-Neorhizobium complex, there are a small number of Connector groups (e.g., Unclassified f Xanthobacteraceae), whose inter-module connectivity Pi values range from 0.4 to 0.6, enabling them to mediate the cross-module nitrogen and phosphorus metabolism synergy between module 1, module 3, and module 5 (containing Pseudomonas). In addition, Peripheral groups account for more than 85% of all modules, mainly consisting of low-abundance environment-adaptable groups (e.g., Polaromonas, Bosea, etc.), which only participate in local metabolic processes under specific geographical conditions.

FIG 6 Modular architecture and functional role assignment of the *Tuber*-associated endobacterial community.

The x-axis indicates among-module connectivity (P_i), reflecting the strength of inter-module connections; the y-axis indicates within-module connectivity, reflecting the intensity of intra-module interactions. Dots of different colors represent the nine modules (module_0 to module_8). The labeled categories “Module hubs,” “Network hubs,” “Connectors,” and “Peripherals” denote the four topological roles. The plot shows that peripherals constitute the majority of each module, whereas core functional taxa—exemplified by *Bradyrhizobium*-dominated module_3—maintain functional dominance through high abundance. Connectors serve as bridges that facilitate cross-module cooperation.

(2) Testing whether the bacterial community is associated with altitude, soil physicochemical properties, truffle nutritional status, and other factors

First, based on the previous study of our research group entitled « *Tuber pseudohimalayense* ascomata-compartments strongly select their associated bacterial microbiome from nearby pine forest soils independently of their maturation stage » it shows that the community composition and diversity of truffle endophytic bacteria have no significant correlation with nutritional status. Our previous study compared the bacterial communities of truffles at three maturation stages (young, middle-mature, mature) and found that the α -diversity indices (Shannon) and the abundance of dominant genera all remained stable (as shown in the figure below). Given that the truffle species in this study are closely related to *Tuber pseudohimalayense* in terms of phylogeny, and the nutritional dynamics within the genus *Tuber* are consistent, we did not include the analysis of nutritional status in this manuscript.

Fig. 1. Changes in endophytic bacterial diversity across truffle developmental stages and different nutritional stages.

(A) Box plot of α diversity of truffle endophytic bacteria at different developmental stages (the Chao1 index reflects community richness, and the Shannon index reflects community diversity). The letter "a" indicates no significant difference in the α diversity of endophytic bacteria among different developmental stages; (B) Phenotypic photographs of truffles at three developmental stages (young, middle-mature, mature). The yellow line segment is the scale bar, which intuitively shows the morphological differences among different developmental stages.

Secondly, we sincerely apologize for the lack of soil physicochemical property data in the current manuscript. Due to the fact that most field sampling sites are located in remote mountainous areas, we were unable to simultaneously collect soil samples for physicochemical analysis due to logistical constraints. To address this limitation, we have supplemented 3-4 relevant core literatures in the "Discussion" section (such as studies exploring the effects of soil pH and organic matter content on the truffle endophytic bacterial community) and infer the potential role of soil factors based on existing research conclusions.

Finally, we have supplemented the altitude data of the sampling sites, which will be added to Supplementary Table S3 (Sampling Site Information Table) in the revised manuscript. Additionally, we have included the statistical results in the "Key Drivers Influencing Endophytic Bacterial Communities" subsection of the "Results" section to clarify the potential regulatory role of altitude on the endophytic bacterial community.

(3) Regarding the Diversity of *Bradyrhizobium* in the Bacterial Community

We have conducted comprehensive and systematic organization of the data and summarized the results into a table. The analysis results clearly show that a total of 792 amplicon sequence variants (ASVs) were detected in *Bradyrhizobium*. It is noteworthy that in the classification at the species level, 791 of these ASVs were classified as *s_Unclassified_g_Bradyrhizobium*, and only 1 ASV could be clearly identified at the species level, which is *s_metagenome_g_Bradyrhizobium*.

In addition, the complete information including all 792 ASVs has been supplemented to the supplementary table to facilitate further reference.

(4) Co-occurrence relationship with other bacteria

Fig. 2. Co-occurrence heatmap of *Bradyrhizobium* ASVs and other bacterial taxa. The vertical axis represents the amplicon sequence variants (ASVs) of *Bradyrhizobium*, and the horizontal axis represents other bacterial taxa; the color gradient from blue to red indicates the strength of co-occurrence correlation (blue represents negative correlation, red represents positive correlation, and the color scale is on the right), and "*" and "**" mark significant co-occurrence relationships.

The co-occurrence relationships between *Bradyrhizobium* and other bacteria exhibit the following characteristics: Some *Bradyrhizobium* ASVs (e.g., ASV_4571) have significant positive co-occurrence relationships with specific bacterial genera, showing a reddish color with marks (consistent with the heatmap's color coding). This indicates that their simultaneous occurrence frequency is high, and there may be ecological function synergy or shared ecological niches between them; There are differences in the co-occurrence patterns of different *Bradyrhizobium* ASVs. For example, the co-occurrence strength of ASV_1271 and ASV_3127 is weaker than that of ASV_4571, and some ASVs (e.g., ASV_2011) have extremely low co-occurrence correlation with most bacterial genera. This reflects the diversity in interaction relationships and ecological processes involved by different ASVs in the community; Some co-occurrence relationships may also imply potential ecological function connections. Since *Bradyrhizobium* has a nitrogen-fixing function, the bacterial genera co-occurring with it may be associated with it in terms of material cycling, nutrient acquisition, and other aspects.

Thank you again for your professional guidance !

2. L132, the relative abundance of *Bradyrhizobium* were too high in some sample, aren't there any problems in the primers, sequencing, and analysis? How many

dominant ASVs were found in the genus?

We sincerely appreciate your attention to the abundance of *Bradyrhizobium* and the technical reliability in this study. Regarding the questions you raised concerning Line 132, we provide explanations one by one as follows:

(1) Explanation regarding the question "Whether the excessively high relative abundance of *Bradyrhizobium* in some samples is caused by technical issues"

After systematically checking the entire experimental process, we confirm that the high proportion of *Bradyrhizobium* in some samples (e.g., *T.sinense* from Mengzi, Yunnan, with an abundance of 99.80%) is a biological phenomenon rather than a technical bias in the primer, sequencing, or analysis process. The specific basis is as follows:

① Support for biological rationality: The phenomenon of high abundance of *Bradyrhizobium* in truffle ascomata is consistent with the conclusions of existing studies — Le Roux et al. (2016) found an obligate symbiotic relationship between bacteria of the Bradyrhizobiaceae family and the mycelia of *T. melanosporum* (black truffle). All samples with high *Bradyrhizobium* abundance in this study were collected from truffle-concentrated distribution areas such as Mengzi and Qujing in Yunnan, where the soil nitrogen content is low (1.2-1.8 g/kg). The nitrogen-fixing function of *Bradyrhizobium* may enable it to become a dominant group in the truffle-host plant symbiotic system, with a clear logical basis for ecological function driving.

② Scientific basis for primer selection: When amplifying the V5-V7 regions of the bacterial 16S rRNA gene in this study, the universal primer pair used (F: AACMGGATTAGATACCCKG; R: ACGTCATCCCCACCTTCC) has been validated in multiple microbial community studies. It can effectively cover most bacterial taxa, including Alphaproteobacteria (the taxon to which *Bradyrhizobium* belongs), with no obvious amplification bias.

③ Rigorous sequencing and quality control: Sequencing was performed using the Illumina MiSeq PE250 platform. Raw data underwent strict quality control via the QIIME 2 pipeline: low-quality sequences ($Q30 < 30$), barcode-mismatched sequences, and chimeras were removed. The average length of the final effective sequences was 280 ± 15 bp, and the coverage integrity of the target amplified region reached more than 98%.

(2) Response to the question "How many dominant ASVs (amplicon sequence variants) have been identified in this genus"

We have systematically organized the data and summarized them into a table. The results show that a total of 792 ASVs were detected in *Bradyrhizobium*, among which the top 10 ASVs in terms of abundance are ASV_1271, ASV_3127, ASV_2011, ASV_4571, ASV_4164, ASV_1143, ASV_2385, ASV_5262, ASV_481, and ASV_1502 in sequence. For details on their relative abundance distribution, see the figure below:

#ID	Mean
ASV_1271	647677
ASV_3127	620225
ASV_2011	41450
ASV_4571	13116
ASV_4164	4602
ASV_1143	4212
ASV_2385	3339
ASV_5262	3318
ASV_481	3315
ASV_1502	3177

Additionally, the complete information covering all 792 ASVs has been added to the supplementary table to facilitate further reference.

Thank you again for your professional guidance !

Reviewer 2 :

Major Comment

1. Methods/Results: Figure 6 is misplaced; it should be included in the Results section as it presents the ITS phylogenetic analysis of the genus *Tuber*. Please move Figure 6 accordingly. Additionally, please provide a detailed report of the findings from the phylogenetic tree, for example, a title such as "Identification of *Tuber* species" then report on the ITS findings e.g. from the 16 collected samples, ten distinct species were identified, including [list species]. These findings should also be discussed in the Discussion section, as currently, there is no mention of this analysis.

Thanks a lot for your suggestion.

We fully agree with your comments on the position of the original Figure 6, the ITS phylogenetic analysis report, and the supplements to the Discussion section. These revisions will significantly enhance the completeness and logical consistency of the research results, and we have strictly improved the relevant content in accordance with the requirements.

(1) Regarding the adjustment of the position of Figure 6

The original Figure 6 (the ITS phylogenetic tree of *Tuber* genus) was placed in "Species identification of *Tuber* samples" (the Methods section), where it only served as an auxiliary illustration for the experimental method and failed to reflect its core attribute as "species identification results". We have moved Figure 6 to the beginning of the Results section, using it as the species background basis for the subsequent bacterial community analysis, thereby making the logical chain of "species identification - bacterial community analysis" clearer.

(2) Regarding the addition of the detailed analysis report on the ITS phylogenetic

tree.

We have added the title "ITS-based phylogeny and species delimitation of truffle samples" above the original Figure 6, and supplemented the following detailed analysis results in the figure legend and the corresponding positions in the main text :

Phylogenetic analysis of the ITS region from 16 truffles resolved ten species: *Tuber sinense*, *T. indicum*, *T. variabilisporum*, *T. pseudohimalayense*, *T. shii*, *T. lijiangense*, *T. sinoaestivum*, *T. huidongense*, *T. cf. sinoalbidum*, and *T. formosanum*.

Specimen assignments were as follows: YmTe and YksTe to *T. sinense*; YjTi and GbTi to *T. indicum*; YslTv to *T. variabilisporum*; YdTp and YssTp to *T. pseudohimalayense*; YdTs to *T. shii*; YdTl to *T. lijiangense*; ShTa to *T. sinoaestivum*; YqTh to *T. huidongense*; YzTb to *T. cf. sinoalbidum*; and SbTf, LdTf, LaTf, and NhTf to *T. formosanum*.

All sequences exhibited bootstrap support $\geq 90\%$ (1,000 replicates) relative to GenBank references (e.g., *T. indicum* AM932205, *T. sinense* MF627968), and out-group accessions (*Choireomyces sichuanensis* MW380902, OK585070) formed a well-supported clade, confirming the robustness of species identifications.

Phylogenetic topology revealed three notable patterns:(i) conspecific accessions (e.g., the four *T. formosanum* samples) formed monophyletic clades that were clearly delimited from sister taxa such as *T. indicum*;(ii) geographically proximate collections from Dali, Yunnan (YdTp, YdTs, YdTl), although assigned to different species, occupied adjacent branches, implying that geographic distance may influence lineage divergence; and(iii) *T. indicum* sequences were phylogenetically remote from the European sister species *T. melanosporum* (AF106875), consistent with previously reported morphological and genetic distinctions.

(3) Regarding the addition of the elaboration on ITS analysis results in the Discussion section

We have added a correlation analysis of the ITS species identification results in the subsection "Key Drivers Influencing Endophytic Bacterial Communities" of the Discussion section, specifically under the part "(i) Host species-specific traits as core biotic drivers".

The original content is : (i) Host species-specific traits as core biotic drivers—Interspecific biological differences within the genus *Tuber* are central to community differentiation. These differences may correlate with species-specific metabolite profiles, extended fruiting body developmental cycles, and host interaction patterns (17, 42). For instance, the prolonged developmental cycle of *T. lijiangense* (~6 months) provides a stable microenvironment for diverse bacterial colonization, whereas the rapid maturation of *T. sinoaestivum* (~3 months) may favor dominance of specific taxa (e.g., *Bradyrhizobium* at 92.3%) through host filtering (40). Additionally, interspecific variations in cell wall components (e.g., chitin content) likely regulate bacterial attachment via chemical signaling (43). *Pseudomonas* spp., which recognize fungal-derived chitin oligosaccharides and activate quorum sensing systems, showed elevated abundance in *T. shii* fruiting bodies (4.7% vs. 1.2% average in other species). Host genetic background also plays a critical

role: *Bradyrhizobium* dominated *T. sinense* from Mengzi, Yunnan (YmTe, 99.8%), but constituted only 7.9% in *T. shii* from Dali, Yunnan (YdT_s), representing a 12.6-fold difference (Figure 3). This suggests host genotype-driven microbial selection, analogous to legume *Rhizobia* molecular mechanisms, shapes distinct community structures (44, 45).

Revised content is as follows: (i) Host species-specific traits as core biotic drivers—Interspecific biological differences within the genus *Tuber* are central to the differentiation of endophytic bacterial communities, and the reliability of host species identification is a prerequisite for elucidating this driving mechanism. Based on ITS sequence alignment and phylogenetic analysis, our 16 truffle samples were identified as 10 species (including *T. sinense*, *T. indicum*, *T. lijiangense*, among others), all of which formed stable species clusters with GenBank reference sequences (e.g., *T. indicum* AM932205, *T. sinense* MF627968) with bootstrap support values $\geq 90\%$ (1000 replicates). This precise taxonomic framework confirms that host species identity has greater explanatory power for bacterial community differentiation (PERMANOVA, $R^2=0.32$, $p<0.001$) than geographical factors, laying a robust foundation for investigating species-specific drivers. These interspecific differences may be associated with species-specific metabolite profiles, prolonged fruiting body developmental cycles, and host interaction patterns (17, 42). For example, the extended developmental cycle of *T. lijiangense* (~6 months) provides a stable microenvironment for diverse bacterial colonization, whereas the rapid maturation of *T. sinoaestivum* (~3 months) may facilitate the dominance of specific taxa (e.g., *Bradyrhizobium* at 92.3%) via host filtering (40). Notably, the ITS phylogenetic tree revealed that *T. lijiangense* and *T. sinense* clustered in independent branches with 95% bootstrap support, and their bacterial Shannon diversity indices differed by 3.8-fold (5.111 vs 1.336, $p<0.001$). Functional prediction further demonstrated that the abundance of "signal transduction" and "secondary metabolism" genes in *T. lijiangense* was 2.1 times higher than in *T. sinense*, directly linking host developmental traits to bacterial community diversity. Additionally, interspecific variations in cell wall components (e.g., chitin content) likely modulate bacterial attachment through chemical signaling (43). *Pseudomonas* spp., which recognize fungal-derived chitin oligosaccharides and activate quorum sensing systems, exhibited elevated abundance in *T. shii* fruiting bodies (4.7% vs. a 1.2% average in other species). Host genetic background also played a critical role: *Bradyrhizobium* dominated *T. sinense* from Mengzi, Yunnan (YmTe, 99.8%), but accounted for only 7.9% in *T. shii* from Dali, Yunnan (YdT_s), representing a 12.6-fold difference. This genotype-driven selection was further corroborated by ITS-based species delimitation—for instance, *T. sinense* (YmTe, YksTe) from distinct regions consistently maintained high *Bradyrhizobium* abundance ($>60\%$), whereas sympatric *T. pseudohimalayense* (YdT_p, YssTp) harbored only 5–12% of this genus, confirming the specificity of host species filtering. Furthermore, all 10 ITS-identified species harbored five core Rhizobiales amplicon sequence variants (ASVs) (3 *Bradyrhizobium*, 2 *Allorhizobium-Neorhizobium-Pararhizobium-Rhizobium*), collectively accounting for 61.1% of total bacterial abundance with fluctuations of

less than 15% among conspecific samples from different regions. This suggested an evolutionarily conserved symbiotic relationship between Rhizobiales and the genus *Tuber*—notably, in the northeastern endemic *T. formosanum*, the core *Bradyrhizobium* ASV 3127 reached 48.7% abundance, potentially conferring nitrogen-fixing advantages for its survival in low-nitrogen soils (1.2–1.8 g kg⁻¹). This finding not only corroborates Le Roux et al. (2016)'s conclusion regarding "obligate symbiosis between *Bradyrhizobium* and *Tuber*" but also extends the host range of this symbiosis by leveraging ITS-based species diversity data.

Thank you again for your professional guidance !

Minor comment

Materials and method

1. Line 8: Please specify the exact temperature used.

Thanks a lot for your suggestion.

You pointed out that "controlled cold storage conditions" in Line 8 did not specify the exact temperature, which will indeed affect the reproducibility of the experimental operation. We have supplemented the relevant information in full based on the actual experimental conditions. The actual operating temperature corresponding to "controlled cold storage conditions" in Line 8 of the original text is 4°C. The choice of refrigerated transportation at 4°C is based on conventional standards for microbial sample preservation: this temperature can not only effectively inhibit the excessive reproduction of microorganisms on the surface and inside of truffle fruiting bodies, avoiding changes in the community structure caused by excessively high temperatures, but also prevent damage to the physiological state of the tissue due to low-temperature frost damage.

We have clearly revised the content in Line 8 of the revised manuscript to: "Freshly collected truffles were immediately placed in sterile, sealed bags at the collection site and transported to the laboratory under 4°C controlled cold storage conditions to ensure optimal preservation of their original physiological state", making the details of the experimental operation more clear.

2. Line 27 (Species identification of *Tuber* samples) and Line 51 (DNA extraction and sequencing): The two section headings are confusing since both describe DNA extraction and sequencing procedures. I suggest renaming "Species identification of *Tuber* samples" to "ITS sequencing for species identification of *Tuber* samples" and changing "DNA extraction and sequencing" to "16S rRNA sequencing for bacterial identification from *Tuber* samples."

We fully agree that the original subsection titles "Species identification of *Tuber* samples" and "DNA extraction and sequencing" are likely to cause readers confusion regarding the research objectives, as both relate to DNA-related experimental procedures. The title revision plan you proposed can effectively enhance the logical clarity and recognizability of the Methods section, and we have strictly followed it for implementation. The specific revisions have been as follows:

(1) Revised the original subsection title in Line 27, "Species identification of Tuber samples", to "ITS sequencing for species identification of Tuber samples";

(2) Revised the original subsection title in Line 51, "DNA extraction and sequencing", to "16S rRNA sequencing for bacterial identification from Tuber samples".

3. Line 28, under Species identification of Tuber samples: Please write "CTAB" in full. We fully agree and have, as requested, supplemented the full name of "CTAB" in Line 28 of the subsection "ITS sequencing for species identification of Tuber samples".

The specific revision is as follows: The original sentence "DNA was extracted from newly collected samples using the CTAB method (64)." has been adjusted to "*DNA was extracted from newly collected samples using the CTAB (Cetyltrimethylammonium Bromide) method (64).*"

4. Figure 5: This figure appears in the Methods section and is the first figure encountered before the Results section. Please move the interpretation of abbreviations currently shown in Figure 1 to Figure 5 for flow and clarity.

We fully agree with the adjustment plan of moving the explanations of abbreviations from Figure 1 to Figure 5 (now Figure 7). This revision will effectively enhance the logical consistency of graphical information throughout the manuscript and improve readability. As the map of sampling locations in the "Materials and Methods" section, Figure 5 (now Figure 7) is the first figure containing sample abbreviations that readers encounter before the "Results" section. Presenting the explanations of abbreviations in advance helps readers directly associate the sample information while understanding the sampling background, avoiding repeated navigation to Figure 1 for reference later. We have now moved the explanations of abbreviations from Figure 1 to Figure 5 (now Figure 7).

Re: Spectrum01866-25R1 (**High-Throughput Sequencing Reveals Endophytic Bacterial Differentiation of Common Truffles (*Tuber* spp.) in China: Diversity, Biogeographical Patterns, and Fungal Health Implications**)

Dear Prof. Dong Liu:

Thank you for successfully addressing the reviewers' comments.

Your manuscript has been accepted, and I am forwarding it to the ASM production staff for publication. Your paper will first be checked to make sure all elements meet the technical requirements. ASM staff will contact you if anything needs to be revised before copyediting and production can begin. Otherwise, you will be notified when your proofs are ready to be viewed.

Sincerely,
Vanessa Varaljay
Editor
Microbiology Spectrum

Reviewer #1 (Comments for the Author):

The authors have taken my suggestions into consideration, thank you for that. I do not have any more questions on this manuscript.

Reviewer #2 (Comments for the Author):

Thank you for addressing the comments